# Effect of “ColdArc” WAAM Regime and Arc Torch Weaving on Microstructure and Properties of As-Built and Subtransus Quenched Ti-6Al-4V

**DOI:** 10.3390/ma17102325

**Published:** 2024-05-14

**Authors:** Anna Zykova, Nikolai Savchenko, Aleksandra Nikolaeva, Aleksander Panfilov, Andrey Vorontsov, Vyacheslav Semenchuk, Denis Gurianov, Evgeny Kolubaev, Sergei Tarasov

**Affiliations:** Institute of Strength Physics and Materials Science of the Siberian Branch of the RAS, pr. Academicheskiy, 2/4, Tomsk 634055, Russia; zykovaap@mail.ru (A.Z.); savnick@ispms.ru (N.S.); nikolaeva@ispms.ru (A.N.); alexpl@ispms.ru (A.P.); vav@ispms.ru (A.V.); svm_70@ispms.ru (V.S.); gurianov@ispms.ru (D.G.); eak@ispms.ru (E.K.)

**Keywords:** wire arc additive manufacturing, “coldArc”, deposition strategy, quenching, microstructure, mechanical properties

## Abstract

Defect-free thin-walled samples were built using wire arc additive manufacturing (WAAM) combined with the “coldArc” deposition technique by feeding a Ti-6Al-4V welding wire and using two deposition strategies, namely with and without the welding torch weaving. The microstructures formed in these samples were examined in relation to mechanical characteristics. The arc torch weaving at 1 Hz allowed us to interfere with the epitaxial growth of the β-Ti columnar grains and, thus, obtain them a lower aspect ratio. Upon cooling, the α/α′+β structure was formed inside the former β-Ti grains, and this structure proved to be more uniform as compared to that of the samples built without the weaving. The subtransus quenching of the samples in water did not have any effect on the structure and properties of samples built with the arc torch weaving, whereas a more uniform grain structure was formed in the sample built without weaving. Quenching resulted also in a reduction in the relative elongation by 30% in both cases.

## 1. Introduction

Wire-arc additive manufacturing (WAAM) is a layer-by-layer metal deposition process using wire as a source material and an arc discharge as an energy source [1,2]. This technology is considered economical and efficient, as well as suitable for the fabrication of large-sized metal products [1] due to its manufacturability, relatively high speed of metal deposition, and low cost of equipment compared to other additive manufacturing methods [2]. Titanium alloys are widely used in various industries, such as aerospace, energy, nuclear, and biomedicine due to their excellent combination of strength and ductility [1,3,4,5]. At the same time, the use of traditional subtractive metalworking methods to produce shaped titanium alloy parts means a significant increase in the production costs. The development of additive technologies offers an alternative way for fabricating near net-shape machine parts from using a layer-by-layer method, growing them from a melt pool. At the same time, the epitaxial growth of primary columnar β-Ti grains is observed with almost all additive manufacturing methods, subsequently leading to the formation of large α/β colonies with wide α-laths that contribute to the anisotropy of properties and reduce the strength.

During the WAAM process, solidification always occurs through the growth of primary grains from the melt pool bottom under the conditions of a temperature gradient and heat removal through the substrate [6,7,8,9]. The relevant studies have shown that the primary β-Ti grains of Ti-6Al-4V alloy produced by WAAM are similar to those obtained using laser [10,11,12] and electron beam additive manufacturing methods [13,14,15,16]. Thus, almost all additive methods that utilize fusion of the source materials give a primary cast structure characterized by large columnar β-Ti grains (several millimeters wide and almost the entire height of the product), in which, at low cooling rates, decomposition occurs with the formation of grain-boundary α -Ti grains and α/β colonies, characterized by low strength and high ductility. With a further decrease in the cooling rate or under the influence of cyclic reheating, the β-laths can experience further decomposition with the formation of α/α″ and α/α_2_ structures [17].

Higher cooling rates are achieved in WAAM as compared to those in wire electron beam additive manufacturing that may cause the transformation of α/β structures into α/α′ ones, and such a transformation may increase strength but simultaneously impair the ductility [18,19]. When additively forming a Ti-6Al-4V part, prolonged heating can also lead to high residual stresses [20]. Thus, products obtained via additive methods are characterized by either an insufficient level of strength associated with the formation of large primary β-Ti grains and their decomposition into coarse α/β colonies, or low plasticity due to the formation of α/α′ colonies. It should be added that the in situ refinement of primary β-Ti grains is possible by breaking the dendrites growing in the melted pool by intensifying the liquid metal stirring by means of mechanical vibrations, sonication (cavitation), and mechanical interpass impact treatment. 

Another specificity of the additively manufactured materials is that successive reheating of the as-deposited layers occurs in them, leading to the formation of heat-affected bands [17], in which the as-deposited microstructures are subjected to extra structural and phase transformations such as the β→α/α″ and β→α/α_2_. Mending such a structural inhomogeneity may require post-deposition heat treatment. 

The modification of deposition strategies may be one of the approaches to interfere with the epitaxial growth of primary columnar β-Ti grains. Plasma or welding torch weaving is an effective way of optimizing the geometry of welding beads and increasing the deposition rate in WAAM [21,22]. In particular, using larger weaving amplitudes in combination with a low torch speed and low wavelength allows for the production of walls of larger dimensions [23]. A novel approach to weaving was developed by Ni et al. [24] for WAAM on curved surfaces that allowed for the improvement of their dimensional accuracy. It was also shown [25] that plasma torch weaving had a positive effect on the structure and strength of reheated zones in the multi-pass plasma-transferred arc Fe-Cr-Mo-C coatings. 

Among numerous publications devoted to WAAM on Ti-6Al-4V, there are ones dedicated to studying the effect of different deposition strategies. For instance, Zhou et al. studied the effect of co-directional and reciprocating deposition strategies on structure and mechanical strength, but no weaving was applied in the deposition process [26]. A circular weaving deposition strategy was applied for wire additive manufacturing of Ti-6Al-4V with the use of the cold metal transfer technique [27]. Unfortunately, no data were reported that could shed light on the effect of low-frequency weaving. No literature sources were, however, discovered that reported on a direct study of the effect of arc or plasma torch weaving in WAAM on the structure and characteristics of as-built Ti-6Al-4V.

It may happen, however, that the use of arc torch oscillations in WAAM has positive effect on the structural homogeneity of the WAAM on Ti-6Al-4V. 

Alternative solutions may be microalloying the melt pool to extend the constitutional undercooled zone and, thus, increase the probability of homogeneous grain nucleation [28] or intensifying the heterogeneous grain nucleation by adding refractory inoculation particles [29]. The first two approaches involve increased design complexity, while admixing alloying and inoculation additives is not always desirable from the viewpoint of maintaining the chemical composition. 

Numerous studies have been carried out to identify the mechanisms determining the microstructural evolution during WAAM and post-heat treatment. For example, the microstructure and mechanical properties of Ti-6Al-4V obtained by the WAAM method were studied after applying five types of the subtransus heat treatment (SHT), from temperatures up to 950 °C [30]. It was found that all five SHT types served to improve both the ultimate strength and ductility. The effect of SHT and hot isostatic pressing on the strength and ductility of Ti-6Al-4V after WAAM was investigated [31]. It was found that various thermal processes directly affect the mechanical properties through the dependence of the strength on the α-grain size [31]. However, it was reported [32] that no significant differences in microstructural or mechanical properties were found after the heat treatment of Ti6Al4V in vacuum (at 720 °C for 120 min), in air, or in argon. All the heat-treated specimens demonstrated both yield and tensile strengths that met the requirements of the AMS4928 standard. A number of sources also indicated the refinement of β-grains, the modification of texture, and residual stresses relieving in the WAAM products after rolling under high pressure [6,8] or sonication [33,34].

As mentioned above, the microstructure of the alloy, including the grain size and morphology, is significantly dependent on the thermal history during the WAAM. In other words, if the thermal conditions during deposition are properly controlled, both the desired microstructures and acceptable mechanical properties can be achieved. Varying the cooling rate by means of interpass cooling with compressed CO_2_ showed a slight improvement in tensile strength and a slight decrease in elongation due to the formation of the α′-phase [35,36]. The Ti-6Al-4V wall was obtained by the WAAM with the use of cold metal transfer (CMT) technique and was then subjected to heat treatment [37]. It was shown [37] that the as-built Ti-6Al-4V wall had a microstructure consisting of acicular α′-martensite with a small amount of lamellar α/β colonies and showed acceptable hardness and tensile strength. The martensitic α′-Ti phase can be eliminated by annealing at 900 °C for 4 h and cooling with the furnace. At the same time, the tensile strength of the heat-treated sample was significantly lower than the tensile strength of the original sample [37].

Thus, thermal post-processing of an additively manufactured sample is a realistic route to provide the desired combination of strength and ductility, but it will require additional costs and time. However, it may be more expedient to control the product manufacturing process using techniques that ensure strictly controlled heat input, such as, for example, “coldArc”, which uses a power source with real time control of the arc discharge ignition and metal transfer [38,39]. The interplay between WAAM process parameters and deposition strategies which utilize the arc (plasma) torch oscillations is very complicated and multifactorial [23,40]. The use of special WAAM deposition techniques such as CMN or “coldArc” makes this relationship even more complicated, but it still needs further investigations. 

In relation to the above-discussed and other papers about using the WAAM process on Ti6Al4V, this work combined WAAM deposition techniques such as “coldArc” with arc torch weaving and evaluated their effect on the grain structure. Thermal post-processing of the samples was carried out to study the necessity of such a procedure for samples obtained by the above-described methods, as well as for understanding the evolution of the microstructure and mechanical properties. 

The objective of this study was to elucidate the effect of arc torch weaving on the microstructure and characteristics of the WAAM Ti-6Al-4V samples obtained with the use of “coldArc” technology.

## 2. Materials and Methods

Welding wire of Ti-6Al-4V, with a diameter of 1.6 mm, was used with the following chemical composition, wt.%: 4.1 ± 0.5 Al, 3.19 ± 0.2 V, Ti—balance. The wire was melted by the arc discharge in a flow of shielding gas (He 99.8%) and then transferred into a melted pool on a 2.5 mm thick rectangular substrate made of commercial pure titanium. The wire arch additive manufacturing (WAAM) was carried out using a 6-axis industrial robot FANUC Robot ARC Mate 100iD (FANUC Europe Corporation, Luxemburg] with a welding torch and an EWM Titan XQ R 400 Pulse inverter power supply (EWM GmbH, Mundersbach, Germany) with “coldArc” technology. The WAAM parameters are presented in Table 1.

The “coldArc” technology allows for the welding process to be carried out with the reduced heat input due to the in situ control of the arc ignition process [41]. It was not possible to calculate the actual heat input due to the lack of parameters and characteristic times of short-circuit and arcing modes. Thus, the nominal value of heat input was calculated the way it is for a conventional arc welding. It may be unnecessary to note here that such an assumption cannot have any effect on the experimental results. Using the WAAM method, two model walls were built from the Ti-6A-4V wire, using different layer deposition strategies: Ti-6Al-4V-0 and Ti-6Al-4V-1 samples (Table 2) were obtained without and with the arc torch weaving, respectively. The weaving amplitude and frequency were 2 mm and 1 Hz (Figure 1). To minimize the distortion of the wall’s geometry, the welding torch was turned in the opposite direction after the deposition of each layer. The angle of inclination of the torch relative to the substrate was 10°. Then, the welding torch was raised to the height of the deposited layer, after which the printing direction was changed to the opposite (Figure 1). The chemical composition of the obtained samples is presented in Table 2.

The as-built walls were sectioned into samples that were subjected to the traditional sample preparation procedure, including the stages of grinding with sandpaper (Al_2_O_3_) and polishing with diamond pastes (14/10, 3/2, and 1/0 grit). To identify microstructure components, the polished surfaces of the samples were subjected to chemical etching with the reagents 2.5 HNO_3_ + 1 HCl + 1 HF + 95 mL H_2_O and 4 HF + 6 HCl + 8 HNO_3_ + 82 mL H_2_O. The macro- and microstructure of the samples were studied using optical microscopy (OM, Altami Met 1S microscope (Altami Ltd., Saint-Petersburg, Russia) and Axiovert 200MAT inverted reflected light microscope(Carl Zeiss AG, Oberkochen, Germany) and scanning electron microscopy (SEM, Thermo Fisher Scientific Apreo S LoVac (Thermo Fisher Scientific, Waltham, MA, USA),) microscope equipped with an energy dispersion spectrometer (EDS)). X-ray diffraction analysis (XRD) was carried out in three zones (bottom, middle, and top) of each wall, using a DRON-7 diffractometer (Bourevestnik, Saint-Petersburg, Russia), Co_Kα_ radiation. The level of microstrains (*ε*) of the WAAM Ti-6Al-4V samples both before and after heat treatment was calculated according to the Williamson–Hall method [42]:(1)$$\varepsilon =\frac{\beta_{hkl}}{4tg\theta }$$
where *β_hkl_* is the full width at half maximum (FWHM), and *θ* is the Wulf-Bragg angle. 

Thin foils were prepared from the thin samples cut from the middle portion of each wall by focused ion beam thinning and then examined by means of a transmission electron microscope (TEM), JEOL-210 (JEOL Ltd., Tokyo, Japan). Uniaxial tensile tests of the dog-bone specimens cut off the corresponding parts of the wall were carried out on a universal testing machine, UTS-110M-100 (Testsystems, Ivanovo, Russia), at room temperature, with a grip speed of 1 mm/min. For uniaxial tensile testing, flat samples were cut with their tensile axes parallel to the OY and with the gauge length dimensions 12 × 2.7 × 1.5 mm.

## 3. Results

### 3.1. Macro-, Microstructure, and Phase Composition of the WAAM Ti-6Al-4V Samples

To study structural evolution of the deposited metal as depended on their thermal history, the walls were arbitrarily divided into three parts designated as bottom, middle and top (Figure 2a and Figure 3a). The macrostructures of all the Ti-6Al-4V samples obtained using “coldArc” WAAM and both layer deposition strategies are represented by former columnar-equiaxed primary β-grains, within which the α-phase exists in the form of lathes and Widmanstätten structures (Figure 2b–d and Figure 3b–d). 

It is worth noting that the primary columnar β-Ti grains are characterized by a relatively small mean length (*d_a_*_m_ = 6 mm) compared to those obtained by WAAM without the “coldArc” technology [1] and other additive methods, where these primary columnar β-grains can reach the length of tens of millimeters and pass through the entire cross-section of the product [11,15,43,44]. The mean size of equiaxed β-grains is almost identical for both layer deposition strategies, i.e., 0.4 ± 0.1 mm and 0.3 ± 0.1 mm in the Ti-6Al-4V-0 and Ti-6Al-4V-1 samples, respectively (Figure 4).

However, the columnar β-grains in Ti-6Al-4V-0 (no weaving) are slightly different in their dimensions, as viewed along the wall’s height. The bottom part of the wall contains the columnar grains with a minimum length and width of 0.57 mm and 0.15 mm, respectively, and a maximum of 5.38 mm and 1.37 mm, respectively (Figure 4a). In the middle part of the sample, some columnar grains can reach their maximum sizes of 8 mm in length and 2 mm in width, which leads to an increase in the mean l/d aspect ratio, as well as to the greater scattering of these values (Figure 4a). The top part of the Ti-6Al-4V-0 wall is characterized by isolated high-aspect-ratio grains with a maximum length of up to 7 mm and a width of 1.7 mm.

The use of 1 Hz arc torch transverse oscillations when depositing the Ti-6Al-4V-1 wall resulted in the formation of more uniform grain structures (Figure 3a). The aspect ratio values in the lower, middle, and upper parts of the wall were 3.4 ± 1.2, 2.7 ± 0.7, and 2.6 ± 0.5, respectively; that is, the values were close to each other and fell within the error limits (Figure 4b). In this case, the maximum length and width of the columnar grains were ~9 mm and ~3 mm, respectively.

The macro- and micrographs of the deposited layers in both the Ti-6Al-4V-0 and Ti-6Al-4V-1 samples demonstrate the presence of large primary α-laths, as well as α/β and α/α′ colonies (Figure 5b,c,g,h). Dark equal thickness bands are observed between the deposited layers in all samples that have been formed as a result of the remelting and reheating of the unmelted part of the layer (heat-affected zone) (Figure 5a,f). The microstructure of the reheating zones differs from that of the deposited layer and is characterized by a larger proportion of primary α-laths and α/β colonies located between them (Figure 5d,e,i,j). However, these heat-affected bands look different in the Ti-6Al-4V-0 and Ti-6Al-4V-1 samples. Weaving allowed us to obtain smaller α/β and α/α′ colonies in the heat-affected band (Figure 5i,j) as compared to those in the Ti-6Al-4V-0 sample (Figure 5d,e).

Figure 6 shows the bright-field TEM images of the microstructures in the Ti-6Al-4V-0 and Ti-6Al-4V-1 samples that look identically as well as contain the same structural components such as α-laths, α/β, and α/α′ colonies. 

The X-ray diffraction patterns of Ti-6Al-4V-0 demonstrate the main reflections from α/α′ (HCP) structures, along with weak reflection from (110)_β_ (BCC), and irrespective of the layer deposition strategy. Since α and α′ have similar crystal structures and lattice parameters, their X-ray diffraction peaks are indistinguishable, and therefore these structures are designated as α/α′ in corresponding X-ray diffraction patterns (Figure 7). 

No reflections were observed from other phases, such as, for example, orthorhombic α″ martensite, which appears in the heat-affected bands after wire electron beam additive manufacturing [17]. The XRD patterns of the Ti-6Al-4V-0 sample demonstrate an inverse order of peak intensities that suggests the existence of some texturing (Figure 7a). The (101)α/α′ peak has the maximum height in the bottom part of the wall, while both (102)α/α′ and (110)α/α′ are the predominant ones in the top part of the wall (Figure 7a). For the Ti-6Al-4V-1 sample, peaks (101) of the α/α′-phase stay identical in regard to height throughout the entire height of the wall, which also indirectly confirms the more uniform structure of this sample compared to that of the Ti-6Al-4V-0 sample (Figure 7b).

### 3.2. The Influence of Heat Treatment on the Structural-Phase State of the “coldArc” WAAM Ti-6Al-4V Samples

The quenching of the Ti-6Al-4V samples from 900 °C followed by cooling in water did not provide any significant microstructural changes (Figure 8). The macrostructure of the WAAM Ti-6Al-4V samples obtained using both layer deposition strategies consisted of columnar-equiaxed primary β-grain boundaries within which the α-phase acquired lamellar, needle-like, and Widmanstätten structures (Figure 8). However, after the subtransus quenching, an increase in the volume fraction of the α′ phase was observed. The mean size of equiaxed grains did not change and was identical to that of the as-built samples; for samples Ti-6Al-4V-0 and Ti-6Al-4V-1, the corresponding mean grain sizes are 0.4 ± 0.1 mm and 0.3 ± 0.1 mm, respectively (Figure 9).

The quenching of the Ti-6Al-4V-0 samples allowed us to obtain more uniform aspect ratio values relating to the primary β-Ti grains (Figure 9a). The mean aspect ratio (l/d) values calculated for the bottom, middle, and top parts of the wall are 3.4 ± 1.1, 3.4 ± 1.1, and 3.6 ± 1.3, respectively (Figure 9a). For the as-quenched Ti-6Al-4V-1 samples, the average aspect ratio values did not change compared to those of the as-built ones, and they were 2.6 ± 0.5, 2.8 ± 0.7, and 3.4 ± 1.2 (Figure 9b).

According to TEM (Figure 10a,b,d,e)), the volume fraction of α′-Ti increased significantly after quenching in both the Ti-6Al-4V-0 and Ti-6Al-4V-1 samples (Figure 10c,f). At the same time, the β-phase was partially retained between the α-plates in the as-quenched Ti-6Al-4V-1 samples (Figure 10e), as confirmed by the corresponding XRD pattern (Figure 11).

The quenching of Ti-6Al-4V-0 samples resulted in the redistribution of the XRD peak intensities compared to those of the as-built samples so that a predominant orientation is observed for the (101)α/α′ reflections (Figure 11a). For the as-quenched Ti-6Al-4V-1 samples, the X-ray diffraction patterns were identical to those obtained on the as-built ones with a predominant orientation of (101)α/α′ over the entire height of the product. 

However, the magnitude of the elastic microstrains of the α/α′ lattice depended on the height of the wall (Figure 12). The as-built Ti-6Al-4V-0 samples demonstrated that their elastic microstrain values increased with the wall’s height. The quenching of the Ti-6Al-4V-0 samples allowed us to decrease their elastic microstrains, but they retained they dependence on the wall’s height (Figure 12a). The as-quenched Ti-6Al-4V-1 samples did not show any linear dependence on the wall’s height, thus showing a maximum in the middle part of the wall (Figure 12b).

### 3.3. Microhardness

The microhardness profiles for the as-built and as-quenched Ti-6Al-4V-0 and Ti-6Al-4V-1 samples allowed us to observe that both the as-built and as-quenched Ti-6Al-4V-0 samples were characterized by the microhardness number scatter, which was greater than that found in the case of the Ti-6Al-4V-1 sample (Figure 13a,b). The average value of microhardness of the as-built deposited Ti-6Al-4V-0 was 3.4 GPa, which then increased to 3.6 GPa after quenching (Figure 13a). For Ti-6Al-4V-1 samples, the microhardness numbers were slightly lower than those of Ti-6Al-4V-0 and had a smaller scatter (Figure 13b). The average microhardness values of the as-built and as-quenched samples were 3.1 GPa and 3.4 GPa, respectively (Figure 13b). The increase in microhardness values after quenching for both layer deposition strategies may be due to the increased volume fraction of the α′-Ti phase.

### 3.4. Mechanical Strength

Tensile stress–strain curves of as-quenched Ti-6Al-4V-0 and Ti-6Al-4V-1 samples obtained without and with the arc torch weaving show that their mechanical characteristics, such as yield strength (YS), ultimate tensile strength (UTS), and relative elongation (**ε**), depend on the wall’s height (Figure 14). The yield strength values for Ti-6Al-4V-0 in the bottom, middle, and top parts of the product are 531 ± 17, 701 ± 20, and 725 ± 21 MPa, respectively; the same characteristics of the Ti-6Al-4V-1 samples are 560 ± 18, 770 ± 21, and 837 ± 25 MPa (Figure 14a). The bottom parts of the Ti-6Al-4V-0 walls demonstrate their UTS and **ε** at the levels of 698 ± 19 MPa and 18%, respectively, while for the Ti-6Al-4V-1 samples, they are 643 ± 18 MPa and 14%, respectively (Figure 14a). The UTS values in the middle and upper parts of the as-quenched Ti-6Al-4V-0 and Ti-6Al-4V-1 samples are identical and amount up to 903 ± 28 MPa. In this case, the relative elongation values in the middle and upper parts of the Ti-6Al-4V-0 wall are 11 and 18%, respectively, as compared to the 10 and 8%, respectively, measured on the Ti-6Al-4V-1 samples. Quenching allowed us to increase the tensile strength in the bottom part of both the Ti-6Al-4V-0 and Ti-6Al-4V-1 samples up to 780–800 MPa (Figure 14b). At the same time, the UTS values in the middle and upper parts did not change and were at the level of ~920 MPa, while the relative elongation was reduced to 5% (Figure 14b).

## 4. Discussion

The Ti-6Al-4V-1 samples were built with the use of arc torch weaving that resulted in smaller β-Ti grains and a more uniform grain structure compared to those of Ti-6Al-4V-0 samples obtained without the weaving. The epitaxial growth of the primary β-Ti grains in the melt pool is determined by the temperature gradient and the associated change in solidification conditions [45,46]. The temperature gradient, nucleation, and growth rate of β-Ti grains are significantly affected by heat input during the additive manufacturing, with a higher temperature gradient typically following less heat input near the liquidus line. Such conditions at the solidification front are accompanied by higher cooling rates, as shown elsewhere [47].

Excessive heat input and possible overheating are known to be problems affecting the stability of the wire arc additive manufacturing process [48,49]. The “coldArc” WAAM features control over the metal transfer via the rapid cycling between arcing and short-circuit modes that requires instantaneous power measurements at the end of each short-circuit mode and immediately before the arc ignition [41]. This ensures a smooth transfer process with a significantly reduced energy input compared to the conventional WAAM. Consequently, the wire arc deposition of the Ti-6Al-4V alloy assisted with “coldArc” technique occurs due to the solidification of a relatively “cold” melt pool, which limits the imposed temperature gradient and, consequently, the columnar growth of the initial β-Ti grains. However, it can be argued that the high heat capacity and low thermal conductivity of the Ti-6Al-4V alloy can lead to significant heat accumulation during the sequential deposition of layers, as well as to the long-term reheating of the previously deposited layers to the temperatures corresponding to the α→β phase transformation, thus making them capable of supporting the epitaxial growth of β-Ti grains.

Deliberate changes in layer deposition speed and the use of low-frequency transverse arc torch oscillations may interfere with the epitaxial grain growth and promote the formation of a more uniform structure with columnar grains of lower aspect ratio as compared to those solidified without using these oscillations.

According to [49,50], excessively high layer deposition speed can lead to insufficient melting of the wire, increase the nucleation center number in the melt pool and thus promote the formation of smaller equiaxed grains in the neighborhood of the columnar ones. However, this was not observed in the present experiment. On the other hand, combined action of oscillations and use of “coldArc” and CMT processes may contribute to extra grain nucleation centers [51].

The use of forced oscillations with a frequency close to that of the natural frequency of the melt pool (10–200 Hz) may cause a resonance phenomenon in the melt pool with corresponding improvements in stirring the melted pool metal [52,53,54]. In addition, the continuous oscillations transmitted via a wire feed mechanism may further improve the stirring of the melt pool, promote fragmentation of dendrites, and grain refinement in the dendrite mushy zone [55]. Fragments of dendrites and grains remain in the melt pool and serve as centers of the β-Ti grain nucleation to provide a competitive grain growth mechanism.

The microstructural inhomogeneities discussed above can affect the mechanical characteristics of the WAAM Ti-6Al-4V. In the titanium α/β alloys, the effective dislocation glide length is determined by the average size of α-colonies and the size of α-laths in Widmanstätten structures [56]. Reducing the size of α-laths and α-colonies will increase the amount of high angle boundaries which serve as dislocation barriers and thus increase the dislocation density along the α/β interfaces that will contribute to improvements of the yield strength and hardness. Therefore, as the size of α-lath decreases, an increase in hardness can be expected. However, the α-lath sizes may be determined for both basket-weave and lath-like microstructures, while α-colonies can allow dislocations to easily glide over significant distances and thus reduce the strength properties of the material [57]. These two features can compensate each other and lead to minimal deviations from average hardness values (Figure 9).

This study demonstrates the effect of using the “coldArc” regime in the WAAM on the Ti-6Al-4V on the microstructure and mechanical characteristics that allowed us to obtain the maximum strength and ductility of the alloy compared to those known from the literature. The obtained UTS is higher than that recommended by the ASTM F1108 standard for deformable Ti-6Al-4V (UTS > 860 MPa, Figure 11). This, along with the process’s ability to reduce structural and mechanical anisotropy, makes the “coldArc” technology the preferred method for the additive manufacturing on Ti-6Al-4V and other Ti alloys. At the same time, “coldArc” for Ti-6Al-4V requires subsequent heat treatment and cannot be used immediately after manufacturing, because the anisotropy of mechanical properties is not eliminated completely even with the use of welding torch weaving (Figure 11). Considering the current mechanical characteristics of the WAAM Ti-6Al-4V-1, the “coldArc” process combined with the arc torch weaving has great potential for additive manufacturing on this alloy. However, further research is required to optimize the process and develop ideal pre- and post-processing routes for the final product.

## 5. Conclusions

In the present study, defect-free WAAM walls were grown from Ti-6Al-4V alloy with the use of the “coldArc” technology and welding torch weaving. The microstructural features of the resulting products were investigated to establish the “process–microstructure–property” relationships. Based on the results obtained, the following conclusions were made: 

It was shown that “coldArc” WAAM with transverse 1 Hz oscillations of a welding torch resulted in the formation of a more uniform grain structure over the entire height of the wall. This effect may be related to the improved melted pool stirring and agitation, which is the dominant factor in achieving dendrite fragmentation and grain refinement during solidification. The use of weaving in combination with the “coldArc” technique allowed us to reduce the aspect ratio of the primary β-Ti grains in the as-built walls, especially when examining the middle part of the wall. This finding was expected from considering the fact that heat conductivity of the top part of the wall was lower as compared to that of the middle part. 

Quenching the samples in water from 900 °C led to the formation of a more uniform grain structure in the Ti-6Al-4V-0 samples built without the welding torch weaving mode. The microstructure (grain size, phase composition, orientation, etc.) and mechanical properties become identical to those found in the as-built Ti-6Al-4V-1 samples obtained with the welding torch weaving.

Quenching had virtually no effect on the structure and properties of samples obtained with the “coldArc” WAAM weaving mode. 

After quenching, the relative elongation decreased in both cases. It has been shown that the use of the “coldArc” technology is promising for the additive manufacturing and high-performance production of Ti-6Al-4V products with a high level of mechanical properties if mechanical anisotropy has been eliminated.

## Figures and Tables

**Figure 1 materials-17-02325-f001:**
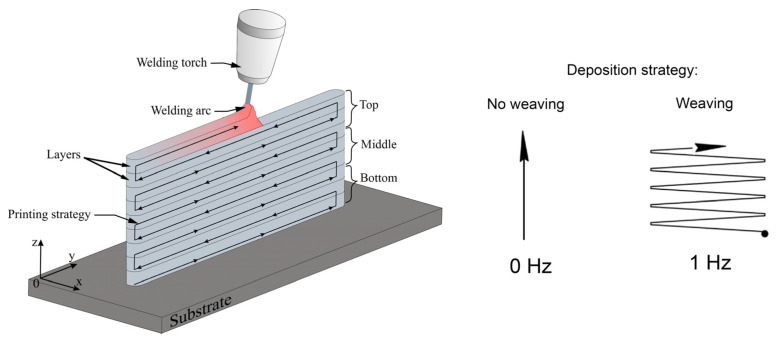
Schematic representation of “coldArc” WAAM on Ti-6Al-4V combined with no weaving/weaving layer deposition strategies.

**Figure 2 materials-17-02325-f002:**
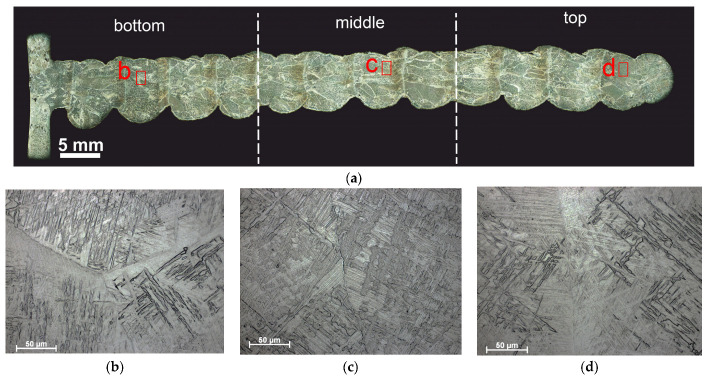
Macrostructure of the sample Ti-6Al-4V-0 obtained via “coldArc” WAAM (no weaving), as-viewed in the ZOX cross-section (**a**), and enlarged optical images of the microstructures obtained in the lower (**b**), middle (**c**), and upper parts (**d**) of the wall. Red letters in (**a**) indicate zones where higher magnification images have been obtained.

**Figure 3 materials-17-02325-f003:**
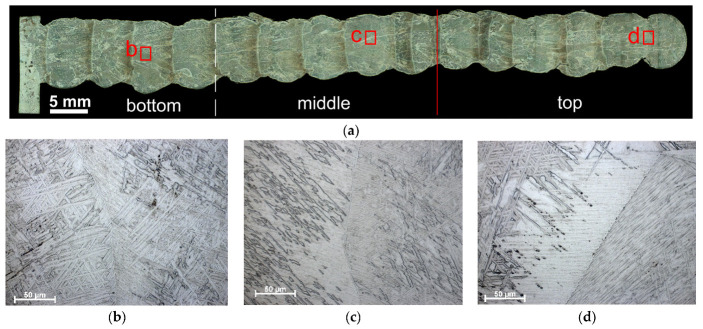
Macrostructure of the sample Ti-6Al-4V-1 obtained via “coldArc” WAAM (no weaving), as viewed in the ZOX cross-section (**a**), and enlarged optical images of the microstructures obtained in the lower (**b**), middle (**c**), and upper parts (**d**) of the wall. Red letters in (**a**) indicate zones where higher magnification images have been obtained.

**Figure 4 materials-17-02325-f004:**
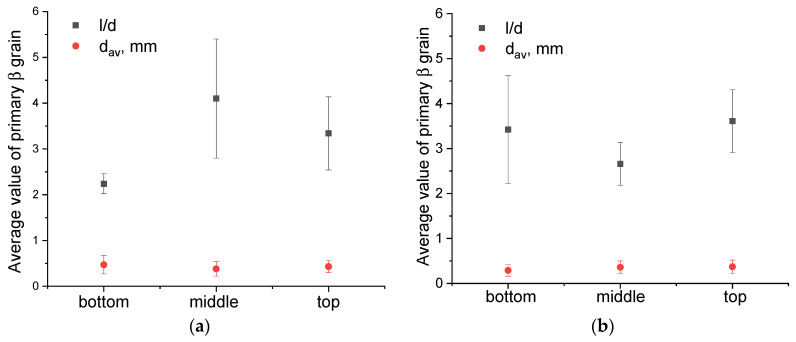
Mean aspect ratio values for primary columnar and equiaxed β-Ti grains in Ti-6Al-4V-0 (**a**) and Ti-6Al-4V-1 (**b**) samples. l/d—mean aspect ratio of primary columnar β-grains; d_av_—mean size of equiaxed primary β-grains.

**Figure 5 materials-17-02325-f005:**
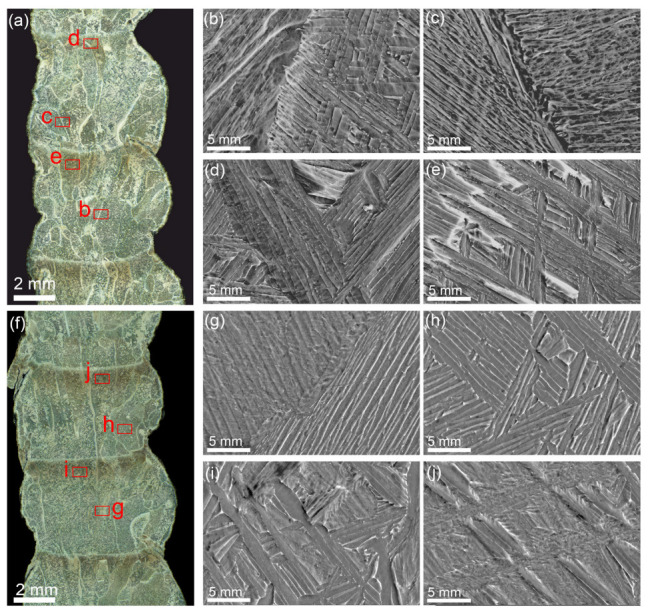
Optical macrographs (**a**,**f**) and SEM images of microstructures in Ti-6Al-4V-0 (**b**–**e**) and Ti-6Al-4V-1 (**g**–**j**). Red letters in (**a**,**f**) indicate areas where higher magnification SEM images have been obtained.

**Figure 6 materials-17-02325-f006:**
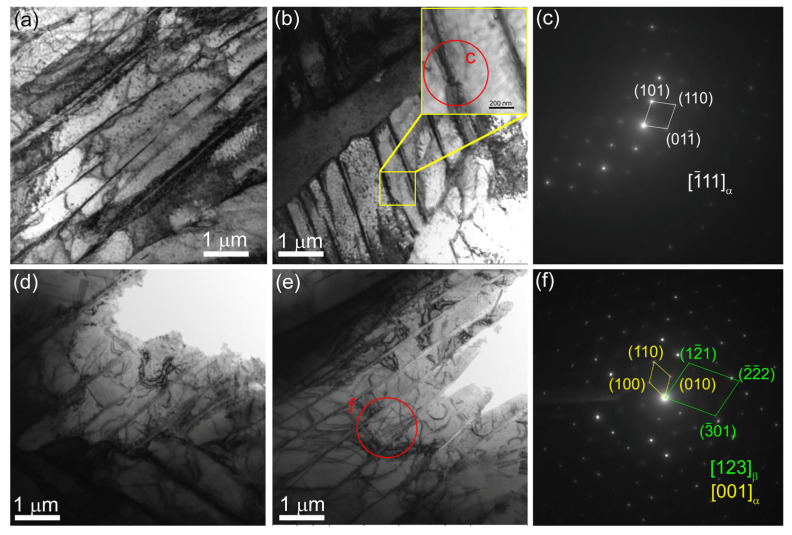
TEM bright-field images (**a**,**b**) and corresponding SAED patterns (**c**) obtained from Ti-6Al-4V-0 (**a**–**c**) and Ti-6Al-4V-1 (**d**–**f**) zones. Red letters and circles indicate the areas from which the SAED patterns were obtained and then identified.

**Figure 7 materials-17-02325-f007:**
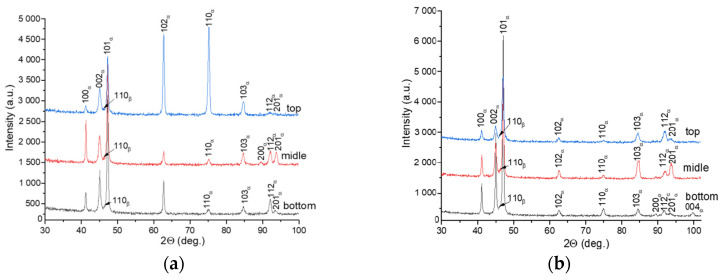
X-ray diffraction patterns obtained from bottom, middle, and top parts of the Ti-6Al-4V-0 (**a**) and Ti-6Al-4V-1 (**b**) walls.

**Figure 8 materials-17-02325-f008:**
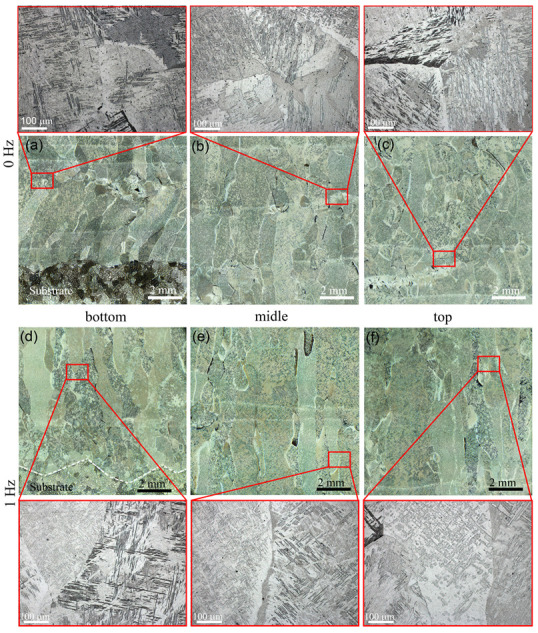
Macrostructures of as-quenched Ti-6Al-4V-0 (**a**–**f**) and Ti-6Al-4V-1 (**b**–**e**) walls, as viewed in a section parallel to the ZOX plane.

**Figure 9 materials-17-02325-f009:**
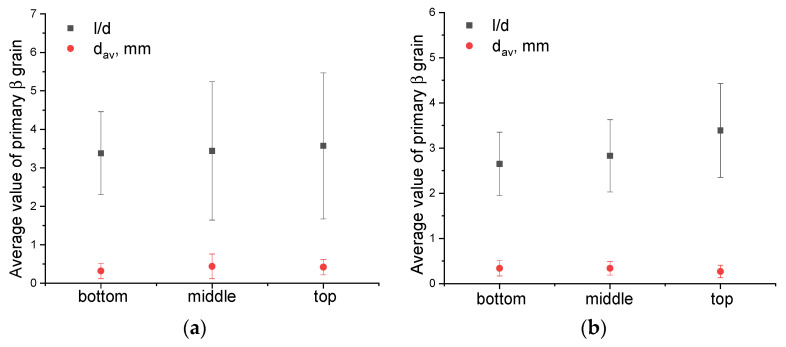
Mean aspect ratio values for primary columnar and size of equiaxial β-Ti grains (d_av_): Ti-6Al-4V-0 (**a**) and Ti-6Al-4V-1 (**b**) as-quenched samples. l/d—mean aspect ratio of primary columnar β-grains; d_av_—mean size of equiaxed primary β-grains.

**Figure 10 materials-17-02325-f010:**
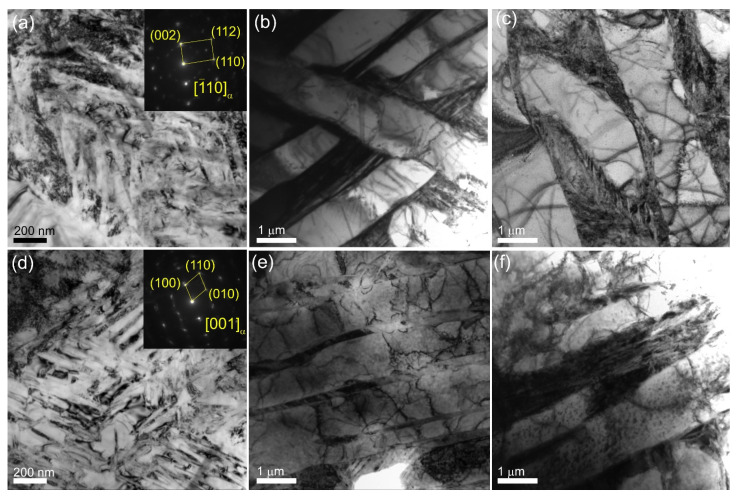
TEM bright-field images of microstructures in as-quenched Ti-6Al-4V-0 (**a**–**c**) and Ti-6Al-4V-1 (**d**–**f**) samples.

**Figure 11 materials-17-02325-f011:**
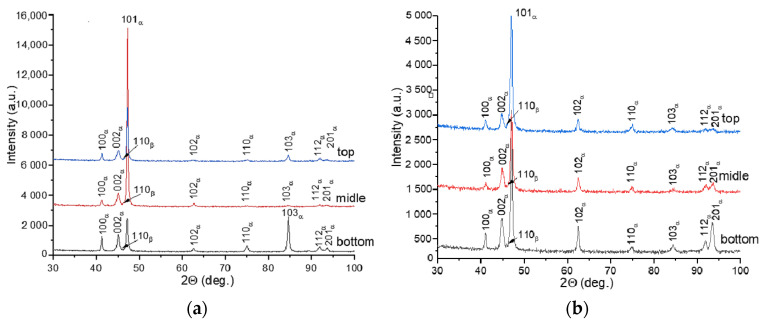
X-ray diffractograms obtained from the as-quenched Ti-6Al-4V-0 (**a**) and Ti-6Al-4V-1 (**b**) samples.

**Figure 12 materials-17-02325-f012:**
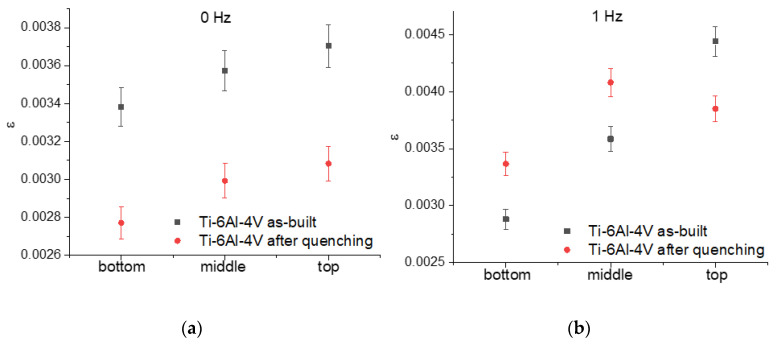
Elastic microstrains on the as-built and as-quenched Ti-6Al-4V-0 (**a**) and Ti-6Al-4V-1 (**b**) samples.

**Figure 13 materials-17-02325-f013:**
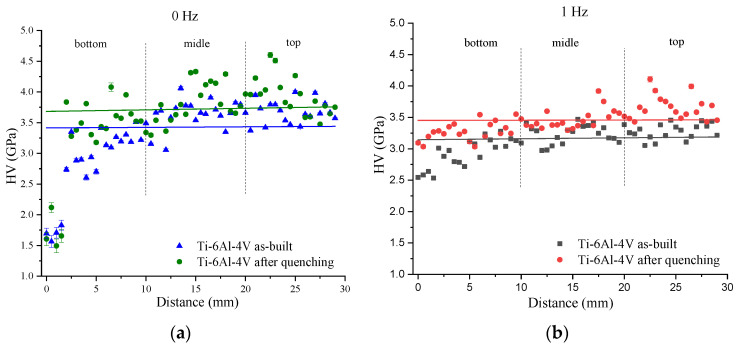
Microhardness profiles of as-built and as-quenched Ti-6Al-4V-0 (**a**) and Ti-6Al-4V-1 (**b**) samples.

**Figure 14 materials-17-02325-f014:**
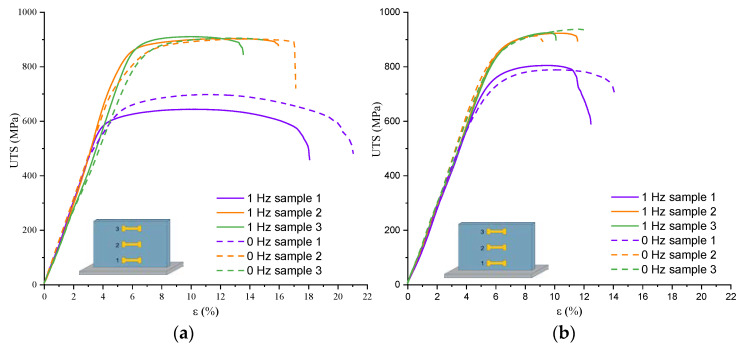
Tensile stress–strain curves obtained from as-quenched Ti-6Al-4V-0 (**a**) and Ti-6Al-4V-1 (**b**) samples.

**Table 1 materials-17-02325-t001:** The “coldArc” WAAM parameters.

Parameter	Magnitude
Mean arc current, A	120
Mean arc voltage, V	14.5
Torch speed, m/min	0.25
Wire feed rate, m/min	3.2
Shielding gas flow rate, l/min	15
Interpass idle time, s	10

**Table 2 materials-17-02325-t002:** Elemental compositions of the “coldArc” WAAM samples.

Layer Deposition Strategy	Zone		Element, wt.%	
		Ti	Al	V
No weaving (Ti-6Al-4V-0)	Bottom	Bal.	3.50 ± 0.57	3.04 ± 0.15
	Middle	Bal.	3.82 ± 0.60	3.09 ± 0.15
	Top	Bal.	4.01 ± 0.62	3.23 ± 0.15
Weaving (Ti-6Al-4V-1)	Bottom	Bal.	3.67 ± 0.56	2.94 ± 0.15
	Middle	Bal.	4.06 ± 0.56	3.03 ± 0.15
	Top	Bal.	4.27 ± 0.63	3.12 ± 0.15

## Data Availability

The data presented in this study are available on request from the corresponding author.

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
