# Peer review of "Effect of “ColdArc” WAAM Regime and Arc Torch Weaving on Microstructure and Properties of As-Built and Subtransus Quenched Ti-6Al-4V"

_materials, 2024, doi:10.3390/ma17102325_

Round 1

Reviewer 1 Report

Comments and Suggestions for Authors

In this work, the microstructure of Ti-6Al-4V samples built by wire arc additive manufacturing with and without torch weaving were compared in details. The authors found that the sample with the weaving given more uniform microstructure with smaller -Ti grain and better mechanical performance. Thermal post-processing of the samples was also carried out to study the evolution of structure and properties. This work might be interesting to the community of WAAM, however, there are in general several issues should be addressed before the final publication.

1.       Arc torch weaving is not new technique, is there any other work about this additive manufacturing method, and how the microstructure like for the samples with similar composition? We all know, the plasma torch weaving is a popular way to improve the mechanical strength of melting or reheated zones of the additive manufactured samples. What the difference between the plasma torch and arc torch weaving/oscillation mentioned in this work?

2.       As shown in Fig. 4 and 9, the grain size is not such different between the one with and without weaving. The authors cannot jump into the conclusion that the one with weaving shows smaller -Ti grains.   

Comments on the Quality of English Language

 There are also some further grammar and spelling mistakes, for example, Lines 276, “emonstrate” should be modified into “demonstrate”.

Author Response

In this work, the microstructure of Ti-6Al-4V samples built by wire arc additive manufacturing with and without torch weaving were compared in details. The authors found that the sample with the weaving given more uniform microstructure with smaller -Ti grain and better mechanical performance. Thermal post-processing of the samples was also carried out to study the evolution of structure and properties. This work might be interesting to the community of WAAM, however, there are in general several issues should be addressed before the final publication.

1.Arc torch weaving is not new technique, is there any other work about this additive manufacturing method, and how the microstructure like for the samples with similar composition? We all know, the plasma torch weaving is a popular way to improve the mechanical strength of melting or reheated zones of the additive manufactured samples. What the difference between the plasma torch and arc torch weaving/oscillation mentioned in this work?

A: Thank you. The majority of literature sources study the effect of weaving on the geometry of the WAAM deposited metal, dimension accuracy, deposition rate etc. Furthermore, we did not manage to find any literature source that dealt with weaving on the WAAM of Ti-6Al-4V for primary grain modification. That is why this work was focused on studying the effect of weaving on the grain structure. The use of specific energy source is not so important because heat input is the main formability process parameter that determines both dimension accuracy and microstructure of any additively manufactured product.  

2.As shown in Fig. 4 and 9, the grain size is not such different between the one with and without weaving. The authors cannot jump into the conclusion that the one with weaving shows smaller -Ti grains.  

A: Thank you. The difference in grain size is very small but still it is. Also the average grain size dav refers only to the equiaxed primary β-Ti grains and not to the columnar ones which are characterized by the average aspect ratio.

Comments on the Quality of English Language

 There are also some further grammar and spelling mistakes, for example, Lines 276, “emonstrate” should be modified into “demonstrate”.

A: Thank you. Corrected.

Reviewer 2 Report

Comments and Suggestions for Authors

-          It is an extremely high-quality paper, with significant scientific and professional achievements.

-          The paper can be accepted, but it is necessary to carry out certain refinements in order to improve the quality of the paper and its application for subsequent tests.

-          It is necessary to review and cite recent paperss from this and similar fields

-          Expand the introduction or other chapters with papers from this field in a significantly larger number.

-          At the end of the introduction, state what is the main contribution of the paper and how does this paper differ from similar papers in this field? What is the reason, ie. why should this papers be published?

-          Increase the number of references in the paper based on the given remarks.

-          you wrote about α, α’, α”, α2 and β particles, but you need to explain more about them, what is their shape, what do they consist of, are they favorable or unfavorable and why?

-          You stated in table 1 cold arc parameters, are they always fixed or they can be changed, if they can be changed why did you select these ones?

-          How did you select waving amplitude and frequency, as well as angle of inclination of the torch?

-          Unify text size on figures for example on figure 13 on a and b text isn’t the same size.

-          What does FWHM mean?

-          In sentence in rows 192 and 193 you stated ”The aspect ratio values  in the lower, middle and upper parts of the wall were 3.4 ± 1.2, 2.7 ± 0.7 and 2.6 ± 0.5,respectively, is there a unit? what are these values?

-          is there porosity in these materials?

-          What is the cost of production? is it good for mass production???

-          It is necessary to avoid starting sentences with abbreviations, as in chapter 2. Materials and Methods.

-          In order to improve the quality of paper, it is necessary to provide key characteristics and pictures or schematic representations of the FANUC Robot ARC Mate 100iD industrial robot and the pulse inverter EWM Titan XQ R 400 in Chapter 2.

-          In Chapter 2, it is necessary to comment to what extent the calculation of the nominal value of the input heat via conventional electric arc welding, instead of the values ​​obtained directly from "coldArc" technology, affects the results obtained and whether it affects the final conclusions presented in the paper.

-          It is necessary to state why the percentage values ​​of mass and element V are not listed in table 2.

-          In chapter 2, it is necessary to state some of the key characteristics of the optical microscope OM Altami Met 1S microscope and Axiovert 200MAT and the electron microscope SEM, Thermo Fisher Scientific Apreo S LoVac, at least in terms of the measurement range, limitations in terms of the working environment, the possibility of visualizing the obtained results, but also display their images or block seeds.

-          Starting from chapter 2, it is necessary to mark the formulas with ordinal numbers in the entire paper, and then to indicate the literature from which the formulas were taken, that is, to indicate whether the authors derived the said formulas independently.

-          On the diagram in Figure 4, it is necessary to mark and better explain in the text which quantities are shown on the x axis of the diagram.

-          It is necessary to state whether the reproducibility of the results was ensured, that is, how many times the experiment was repeated and whether at least approximately the same conditions of the experiment were ensured each time.

-          In the Discussion, it was stated that an excessively high speed of layer application can lead to insufficient melting of the wire, increasing the number of nucleation centers in the melting pool, and thus encouraging the formation of smaller equiaxed grains in the neighborhood of columnar ones, which is explained in earlier literature. In addition, it is stated that this was not observed in this experiment, but it is not stated whether the mentioned phenomenon is not observed for all repetitions of the experiment and all operating conditions.

-          In the concluding remarks, it is necessary to clearly state whether the obtained results confirmed the expectations and assumptions, that is, whether there were deviations or completely opposite results from the expected ones.

-          Based on the analysis and discussion, expand the concluding considerations, especially in terms of further research in the field of application of other materials and coatings and other experimental conditions. Consider the possibilities of commenting on the reduction or increase in the costs of production, exploitation and maintenance of components made of this type of alloy, as well as whether there are any harmful effects of the application of these alloys and materials on the environment and human health. 

Author Response

Reviewer 2

It is an extremely high-quality paper, with significant scientific and professional achievements.The paper can be accepted, but it is necessary to carry out certain refinements in order to improve the quality of the paper and its application for subsequent tests.It is necessary to review and cite recent paperss from this and similar fieldsю Expand the introduction or other chapters with papers from this field in a significantly larger number.

At the end of the introduction, state what is the main contribution of the paper and how does this paper differ from similar papers in this field? What is the reason, ie. why should this papers be published?

A: Text has been added in accordance with the comments

Increase the number of references in the paper based on the given remarks.

A: Thank you. Revised

you wrote about α, α’, α”, α2 and β particles, but you need to explain more about them, what is their shape, what do they consist of, are they favorable or unfavorable and why?

A: Thank you. It is common knowledge that α, α’, and α” phases are HCP phases of Ti, β is the BCC high temperature titanium, α2 is the chemical compound Ti3Al.

 β®α transformation at equilibrium conditions

β®α’ is the martensitic transformation at high cooling rate. phase.

β®α” is the martensitic transformation at medium cooing rate

α2 -Ti3Al is the product of decompoistion from the β-Ti in heating

You stated in table 1 cold arc parameters, are they always fixed or they can be changed, if they can be changed why did you select these ones?

A: Thank you. These parameters were experimentally adjusted in order to provide building a quality wall. Changing these parameters would cause defects.

How did you select waving amplitude and frequency, as well as angle of inclination of the torch?

A: Thank you. These parameters were selected from experimental conditions to obtain the quality walls. The weaving frequency was related with the heat input. 

Unify text size on figures for example on figure 13 on a and b text isn’t the same size.

A: Thank you. Corrected.

What does FWHM mean?

A: Thank you. FWHM is a standard characteristic of an XRD peak broadening and means full width at half maximum.

In sentence in rows 192 and 193 you stated ”The aspect ratio values  in the lower, middle and upper parts of the wall were 3.4 ± 1.2, 2.7 ± 0.7 and 2.6 ± 0.5,respectively, is there a unit? what are these values?

A: Thank you. Since these value represent a ratio between length and width of the grain, and therefore, they are dimensionless.

is there porosity in these materials?

A: The process parameters were experimentally optimized for obtaining fully dense samples.

What is the cost of production? is it good for mass production???

A: We did not evaluated the production costs because we are focused on experimental works. However, this method is counted as the cost-effective one.

It is necessary to avoid starting sentences with abbreviations, as in chapter 2. Materials and Methods.

A: Thank you. Corrected.

In order to improve the quality of paper, it is necessary to provide key characteristics and pictures or schematic representations of the FANUC Robot ARC Mate 100iD industrial robot and the pulse inverter EWM Titan XQ R 400 in Chapter 2.

A: Thank you. We believe it would be unnecessarily to show these details in the research paper All details may be found on the Internet .

In Chapter 2, it is necessary to comment to what extent the calculation of the nominal value of the input heat via conventional electric arc welding, instead of the values ​​obtained directly from "coldArc" technology, affects the results obtained and whether it affects the final conclusions presented in the paper.

A: Thank you. Revised.

It is necessary to state why the percentage values ​​of mass and element V are not listed in table 2.

A: It is common practice to show the chemical composition of the materials under study in wt.%.

In chapter 2, it is necessary to state some of the key characteristics of the optical microscope OM Altami Met 1S microscope and Axiovert 200MAT and the electron microscope SEM, Thermo Fisher Scientific Apreo S LoVac, at least in terms of the measurement range, limitations in terms of the working environment, the possibility of visualizing the obtained results, but also display their images or block seeds.

A: Thank you. We believe it would be unnecessarily to show these details in the research paper. It is also good practice in research papers to indicate the commercial name of the equipment without showing their photos. All the details may be easily found on the Internet.

Starting from chapter 2, it is necessary to mark the formulas with ordinal numbers in the entire paper, and then to indicate the literature from which the formulas were taken, that is, to indicate whether the authors derived the said formulas independently.

A: Thank you. Corrected.

On the diagram in Figure 4, it is necessary to mark and better explain in the text which quantities are shown on the x axis of the diagram.

A: Thank you. The abscissa axis in Fig.4 is a dimensionless one and serves to show which part of the wall was investigated to obtain the mean aspect ratio and mean size of the primary grains. .

It is necessary to state whether the reproducibility of the results was ensured, that is, how many times the experiment was repeated and whether at least approximately the same conditions of the experiment were ensured each time.

A: Thank you. Each experimental datapoint was obtained by averaging over at least three experimental results. The microstructure of the WAAM sample (wall) is fully determined by the heat input. Once determined the process parameters it would easy to reproduce the wall. Otherwise, this method would have never gained such popularity.

In the Discussion, it was stated that an excessively high speed of layer application can lead to insufficient melting of the wire, increasing the number of nucleation centers in the melting pool, and thus encouraging the formation of smaller equiaxed grains in the neighborhood of columnar ones, which is explained in earlier literature. In addition, it is stated that this was not observed in this experiment, but it is not stated whether the mentioned phenomenon is not observed for all repetitions of the experiment and all operating conditions.

A: We did not observe the discussed phenomenon abiding with the process parameters (heat input) previosly optimized for providing the best possible quality of the wall. Let us remind that changing the process parameters would interfere with deposition and introduce defects.

In the concluding remarks, it is necessary to clearly state whether the obtained results confirmed the expectations and assumptions, that is, whether there were deviations or completely opposite results from the expected ones.

A: Thank you. Revised

Based on the analysis and discussion, expand the concluding considerations, especially in terms of further research in the field of application of other materials and coatings and other experimental conditions. Consider the possibilities of commenting on the reduction or increase in the costs of production, exploitation and maintenance of components made of this type of alloy, as well as whether there are any harmful effects of the application of these alloys and materials on the environment and human health. 

A: Thank you. We were focused on studying the effect of WAAM on microstructutres and properties of the Ti6Al4V samples. However, it is known that WAAM is the most popular additive method that requires only welding machine and manipulation arm or table. Therefore it is  so popular among other additive methods. 

Reviewer 3 Report

Comments and Suggestions for Authors

Comments to author

The manuscript is well-written and well-organized, and the subject aligns with the scope of the materials. The authors presented defect-free thin-walled Ti-6Al-4V samples using wire arc additive manufacturing combined with "coldArc" deposition. The article is descriptive and should be of interest as it thoroughly explains the effect of WAAM process and heat treatment on the microstructure and mechanical properties of Ti6Al4V samples. The English and references are satisfactory, but the manuscript needs some revisions.

1. Line 119-120: correct the typo.

2. Figure 2 : It would be helpful to roughly indicate which part of (a) corresponds to the lower, middle, and upper parts depicted in (b), (c), and (d) respectively. Additionally, please adjust the contrast of (b), (c), and (d) slightly darker to enhance the visibility of the brighter areas.

3. Please adjust the text displayed on Figures 5 (a) to (j) to make it more visible.

4. The microstructure images in (b) and (c) of Figure 8 are identical. Please verify this and make the necessary corrections.

5. Please adjust the font size of the text in graphs 4 and 9 as they are too small.

Comments on the Quality of English Language

Minor editing of English language required

Author Response

The manuscript is well-written and well-organized, and the subject aligns with the scope of the materials. The authors presented defect-free thin-walled Ti-6Al-4V samples using wire arc additive manufacturing combined with "coldArc" deposition. The article is descriptive and should be of interest as it thoroughly explains the effect of WAAM process and heat treatment on the microstructure and mechanical properties of Ti6Al4V samples. The English and references are satisfactory, but the manuscript needs some revisions.

1. Line 119-120: correct the typo.

A: Thank you. Corrected.

  1. Figure 2 : It would be helpful to roughly indicate which part of (a) corresponds to the lower, middle, and upper parts depicted in (b), (c), and (d) respectively. Additionally, please adjust the contrast of (b), (c), and (d) slightly darker to enhance the visibility of the brighter areas.

A: Thank you. Corrected.

  1. Please adjust the text displayed on Figures 5 (a) to (j) to make it more visible.

A: Thank you. Corrected.

  1. The microstructure images in (b) and (c) of Figure 8 are identical. Please verify this and make the necessary corrections.

A: Thank you. Corrected.

  1. Please adjust the font size of the text in graphs 4 and 9 as they are too small.

A: Thank you. Corrected.

Comments on the Quality of English Language

Minor editing of English language required

A: Thank you. Corrected.
